# Analyzing robust dividend payout policy with dynamic panel regression: Application of speed of adjustment to half-life

**Kittisak Jangphanish**[1], **Wachira Boonyanet** [2,3]*, **Supa Tongkong**[4]

**1** Faculty of Science and Technology, Rajamangala University of Technology Rattanakosin, Nakhon Pathom, Thailand, **2** Chulalongkorn Business School, Chulalongkorn University, Bangkok, Thailand, **3** Visiting Scholar at Brunel University London, London, United Kingdom, **4** Faculty of Business Administration, Thongsook College, Bangkok, Thailand

* wachira@cbs.chula.ac.th

**Data Availability Statement:** All relevant data are within the paper and its Supporting Information files.

## Abstract

This study aims to observe the effects of financial metrics, free float shareholders, GDP growth and firm age on dividend payout policy in both the short- and long-term scenarios. The study introduces the dynamic panel regression models, i.e. Autoregressive Integrated Moving Average (ARIMA) for time series data and Weighted Least Squares (WLS) to account for autocorrelation and heteroscedasticity limitation. The dataset includes all companies listed on the Stock Exchange of Thailand during the years 2013–2023. The study finds that in the long-term, GDP growth negatively relates to dividend payout policy in all industrial sectors. Financial metrics, free float shareholders, GDP growth and firm age affect a mixed picture of industrial sectors on dividend payout policy. In the short-term, previous dividend payments significantly influence dividend payment policy. Furthermore, a higher debt-to-equity ratio, firm age, and free cashflows influence dividend payout policy in various industrial sectors. Also shown by the analysis is that factors influence short-run adjustment to half-life analysis.

## 1. Introduction

Prior studies on dividend payout policy and its relationship with key indicators have been the subject of intense scrutiny [1]. The nature of the involved factors is not only controversial [2–7] but also has serious implications for ensuring the viability of long-term sustainability. Furthermore, although previous research has focused on listed companies, firms in different industries reveal varying dividend payout patterns based on their unique circumstances. Management of firms should recognize the factors influencing dividend payout policy in particular conditions from time to time, and importantly, manage their organizations for reasons of financial sustainability and the need to ensure long-term growth. Meanwhile, stakeholders should clearly scrutinize the information about the firm to determine whether it benefits them and reflects their interests and investment plans.

**Funding:** The author(s) received no specific funding for this work.

**Competing interests:** The authors have declared that no competing interests exist. Authors have no competing interests.

Although theories such as the dividend signaling theory [8,9] and the value relevance of accounting information theory [10] provide important perspectives, there is ongoing debate regarding which factors most significantly influence dividend payout policies in practice. Previous studies, especially with regard to heteroscedasticity and autocorrelation, reveal a methodological gap in the statistical methods used to investigate the link between dividend payment policy and important elements. Often using less advanced methods, traditional regression models caused the results to be misunderstood [3,7,11]. Particularly in emerging and market-driven nations, operational efficiency parameters like asset turnover, free float shareholders, company age, and GDP growth are often neglected even if financial measures including profitability and earnings have been much researched.

## 1.1 Statement of the problems

The dividend distribution policy and its interaction with important criteria remain understudied and controversial in markets driven by development and in emerging economies. Although most studies now underline financial indicators, they can overlook important operational and macroeconomic factors such as asset turnover, free cashflows, GDP growth, and company age. This discrepancy in the literature compromises our knowledge of dividend distribution policy in many different economic environments and sectors of business both in long-term and short-term perspectives. Furthermore, traditional regression techniques often used in past research neglect important statistical problems such as homoscedasticity and autocorrelation. These methodological flaws provide biased and inconsistent results, consequently reducing the validity of the conclusions derived from such investigations.

The important elements that still need further research are as follows. Based on the different sectors and economic environments, what are the main long-term elements affecting dividend distribution policy? Once long-term elements are discovered, what are the short-term variables of dividend payout policy and how do they interact with one another? Finally, what is the speed of adjustment of companies changing their dividend distribution policies to deviations? These unsolved problems emphasize the requirement of a thorough methodology to examine dividend payment policies, including both long-term and short-term dynamics as well as the speed of adjustment to equilibrium.

## 1.2 Research objectives

Observing the factors influencing dividend payment policy, this study has the following main goals:

1. Determine and evaluate, in developing and market-driven economies, the main long-term elements like asset turnover, free cashflows, GDP growth, company age, and others that greatly affect dividend payment policies using the integrated panel regression and ARIMA approach and Weighted Least Squares (WLS), extending restrictions in past research for robustness conclusions.

2. Examining the short-term elements influencing dividend payout policy, concentrate on how these factors interact with and complement the long-term determinants.

3. Analyze the pace at which companies change their dividend payment policies to deviate from short-term equilibrium in different industries.

By reaching these goals, the research aspires to providing both important theoretical insights and pragmatic suggestions for company managers and legislators in developing and market-driven countries.

This study offers a thorough understanding of how asset turnover, debt-to-equity ratio, free cashflows, net cashflows, free float shareholders, GDP, and firm age influence dividend payout policy over the long-term. The most significant finding is that GDP growth negatively and significantly relates to dividend payout policy. Regarding the short-term scenario, debt-to-equity ratio, firm age, and cashflows affect dividend payout policy. Especially, the past dividend payments significantly influence dividend payment policy. Fixed Effect Regrssion Model (FEM) is the most effective among the Pooled OLS Regression and Random Effect Regression Model (REM). FEM can control external influences without adding bias through dummy variables. However, FEM is still questionable on the issues of autocorrelation and heteroscedasticity (both called endogeneity) because it can shape the reliability of the findings. Endogeneity develops when independent factors correlate with the error term, causing bias. To make the most of this, instrumental variables (IV) are deployed, especially when utilizing ARIMA-type strategies. IVs aid in obtaining consistent estimates by reducing endogeneity. Techniques like Two-Stage Least Squares (2SLS) and Generalized Method of Moments (GMM) are typically used in combination with IVs. Consequently, IVs constitute alternative ways such as the control function method, which is helpful for nonlinear models. Maximum likelihood estimation, despite the fact that it is more efficient, is notable for its difficulties in distribution definition [12–17].

The rest of the paper is structured as follows. Section 2 encapsulates the literature review including comments on the underpinning theories and relevant studies. Section 3 explains the conceptual framework deployed in this analysis. Section 4 outlines the research methodology. Section 5 deals with the findings that emerge in this research. Finally, Section 6 incorporates the discussion and concluding comments on the main themes covered in this paper.

## 2. Literature review

### 2.1 Underpinning theories

This study investigates the factors influencing dividend payout policy by integrating two key theories: the dividend signaling theory and the value relevance of accounting information theory. These theories provide the foundation for: firstly, selecting the variables examined in the empirical model; and secondly, understanding the mechanisms through which these factors influence dividend payout decisions.

The value significance of accounting information theory and the supporting theory of this research including the dividend signaling theory are discussed. Dividend signaling theory holds those changes in dividend distributions function as a management signal regarding predicted future profitability. While an increasing dividend pay-off shows management has the confidence to produce greater future earnings, a declining dividend payout is more likely to suggest expected financial troubles. Variables such as net cashflows, free cashflows, and earnings per share are used in this theoretical framework. Strong cash flow firms are more likely to signal the market in a positive direction through increasing dividends. This study uses free cashflows and net cashflows as proxies for liquidity and earnings potential, therefore capturing the firm's ability to meet operational and investment needs and so facilitate dividend distribution. These elements help the model operationalize the dividend signaling theory, so whether businesses use dividend changes to imply investors' future financial stability and growth potential can be investigated.

The value relevance of accounting information theory asserts that financial data offer analysis for evaluating firm performance and decision-making. Emphasizing accounting information in deciding dividend policies, this theory shows a relationship between financial information and its potential to deliver benefits. In this context, variables like the debt-to-

equity ratio, asset turnover provide value relevance to dividend payout policies. The debt-to-equity ratio reflects financial risk and the potential need to retain earnings to service debt, which may limit dividend payouts. Asset turnover measures operational efficiency, indicating the firm's ability to generate revenue from its assets, which can lead to higher residual earnings available for dividends.

By incorporating these theories, this study seeks to bridge the gap between theoretical insights and empirical findings, offering a robust understanding of the determinants of dividend payout policy.

## 2.2 Relationship of dividend policy-related factors

Normally, the association between various indicators and dividend policy has been identified. **Asset turnover ratio** shows the operational efficiency of asset management which indicates high quality asset management. This leads to higher residual earnings for dividends [18,19]. However, a higher **debt-to-equity ratio** strongly suggests greater financial risk, because the company needs to retain earnings to pay debts and interest on loans instead of paying dividends to shareholders [20–22]. **Free cashflows and net cashflows** are important for business operations as they indicate the availability of liquidity after paying capital expenditure and operating expenses. Therefore, cashflows data is extremely important in deciding dividend payout policy [23–26]. **Free float shareholders** are deemed to be an influential factor affecting dividend policy.

Companies with higher free float shareholders might admit increased pressure to pay dividends regularly and are more likely to invest in those stocks which pay higher dividends to meet their demand [27]. As well, higher **GDP growth** leads to larger dividend payouts. This is because companies with higher sales and profits tend to have greater capacity to pay dividends. However, during times of economic turmoil or industry downturn, companies may adopt more conservative dividend policies to safeguard their liquidity. Finally, companies during their early periods of existence tend to reinvest earnings and pay only small or no dividends [11]. Companies which are **aged** or mature are more likely to enjoy more stable earnings and less growth opportunities, resulting in higher and more consistent dividend payments [28].

## 2.3 Previous studies on dividend policies

The previous literature can be categorized into three main types: emerging markets, developed markets, and sector-specific studies, each providing different insights into the factors affecting dividend policy in different contexts. Emerging market research emphasizes the specific characteristics that influence dividend policy. The Gulf Cooperation Council (GCC) research by [2] supports the traditional dividend theory while providing insights into the specific characteristics of these markets. It concludes that the importance of investment opportunities, liquidity, firm size, and ownership structure shape dividend policy. [3] undertook their study in Indonesia and found that firm size and profitability negatively affect dividend policy, although financial structure has no significant impact on dividend policy studied. [4] replicated this analysis to Vietnam and revealed that the importance of dividend policy serves as a function of operating profit and firm size, while emphasizing the risk associated with high financial policies affecting dividend policy. [5] emphasized both internal factors like profitability, corporate governance and external influences like investor demand guide dividend policy determination. [6] showed that board characteristics such as board size and tenure positively affect dividend payout, while factors, for example, board diversity and the tenure of both the chairman and the executive chairman have only a minor impact in Malaysia. [11] discovered that profit, firm

size and GDP are associated with dividend payment policy. [3,7] also found that profit, firm size, and corporate tax significantly influence dividend policy.

Analyses of the efficiency of markets, previous studies focused on conventional determinants such as earnings, past dividend payment, and industry factors. [29] analyzed NYSE-listed companies and observed that dividend payout policy is influenced by earnings, share price and cash flow. Also, [30] found that dividend payout policy is influenced by current and expected earnings, past dividend payment, and industrial environment influences in companies listed on NASDAQ. Previous research also documents other factors influencing dividend policy. For example, [31] found that microeconomic and macroeconomic factors determine dividend policy in the mining sector in Indonesia. [32] reported that audit quality mediates the relationship between corporate governance and dividend policy and all of these factors influence firm performance. [33] revealed that growth opportunities, profitability, and monetary policy affect dividend payout policy in Ghana. [34] contended that cash position decides dividend payout policy. Lastly, [35] found that industrial-specified factors guide dividend payment policy in India.

**2.3.1 Literature gap.** Notwithstanding that a lot of research has been done on dividend payment rules, there are still notable gaps in the literature. First, especially in the context of developing and market-driven economies, operational efficiency elements like asset turnover, free float shareholders, and firm age have received comparatively little attention despite the fact that financial measures including profitability and earnings have been well investigated. Second, research depends on conventional regression methods that neglect important methodological concerns such as heteroscedasticity and autocorrelation, hence producing biased results. Third, most studies concentrate on developed economies or certain sectors, and therefore lacking knowledge on how dividend payment policies function across sectors in developing countries, defined by different economic conditions and financial systems.

Moreover, especially in times of economic instability, current research usually overlooks the interaction between macroeconomic factors (e.g., GDP growth) and dividend policy. This exclusion restricts our knowledge of how outside economic circumstances affect choices on dividend policy. Finally, even if theoretical models such as the dividend signaling theory and the value relevance of accounting information theory offer a basis, there is little empirical evidence of their application in various economic settings.

**2.3.2 Contribution of the current study.** Focusing on the link between dividend payment policy and important elements such as asset turnover, debt-to-equity ratio, free cashflows, net cashflows, free float shareholders, GDP growth, and company age both long-term and short-term perspective in various sectors, this study seeks to close these gaps. This work addresses methodological constraints in previous studies by using sophisticated statistical approaches including ARIMA and WLS, therefore providing strong and consistent results. Furthermore, the research looks at these links in the framework of developing and market-driven countries, therefore offering information that advances both theoretical and pragmatic knowledge of dividend programs.

## 3. Conceptual framework

The literature review presented above identifies a research opportunity for this study. S1 Fig below is the conceptual framework and it illustrates the internal factors variables such as asset turnover, debt to equity, free cashflows, net cashflows, and firm age, along with the presence of free float shareholders; all wield a direct impact on dividend policy. Furthermore, the external economic environment like GDP growth affects dividend policy. This integrated perspective underscores the complex factors influencing dividend policy.

## 4. Research methodology

### 4.1 Dataset and statistical analysis

The population of this study consists of all companies listed on the Stock Exchange of Thailand (SET). Bloomberg database is selected as the trusted source of data encompassing the years 2013 to 2023. The study classifies 8 industrial sectors of listed companies, and these are: AGRO (Agriculture) 836 observations, CON (consumer products) 682 observations, FINANCE (financial companies) 902 observations, INDUS (Industrial) 1,474 observations, PROP (Property) 2,178, RES (Resources) 847 observations, SER (Services) 2,145 observations, and TECH (Technology), 726 observations. Before analyzing the data, selected financial data was meticulously extracted from the Bloomberg database and compared with SETSMART, the SET official website to avoid data duplication and ensure reliability. The study employs the dynamic panel regression models, i.e. Autoregressive Integrated Moving Average (ARIMA) for time series data and Weighted Least Squares (WLS) to account for autocorrelation and heteroscedasticity limitation.

### 4.2 Measurements for the variables

Table 1 summarizes the comprehensive details on how the study's dependent and independent variables were measured.

### 4.3 Model specifications

In order to achieve the study's aim, the equation is presented here:

$$\text{DIV}_{it} = \beta_0 + \beta_1 AT_{it} + \beta_2 DE_{it} + \beta_3 FCF_{it} + \beta_4 NCF_{it} + \beta_5 FFS_{it} + \beta_6 GDP_{it} + \beta_7 AGE_{it} + \beta_8 DE{*}NCF_{it} + ARIMA(p,d,q) + \varepsilon_{it}$$

The results of the analysis are shown in Table 3.

Then, the analysis undertakes the long run to equilibrium [36] and the models are presented below:

$$\hat{y}_t = \alpha_0 + \beta_1 X_{1,t} + \ldots + B_n X_{n,t} + u_t \tag{1}$$

and $\hat{u}_t = y_t - \alpha_0 - \beta_1 X_{1,t} - \ldots B_n X_{n,t}$, If $\hat{u}_t$ is stationary the variables cointegrated with a long run to equilibrium, so the equation of the long run to equilibrium is as follows:

**Table 1. Measurement of study variables.**

| Variables | Acronym | Measurements | |
|---|---|---|---|
| **Dependent variable** | | | |
| Dividend payout ratio | DIV | Distributed dividend over net income | |
| **Independent variable** | | | |
| Asset turnover | AT | Sales to total assets | |
| Debt to equity | DE | Debt to equity | |
| Free cashflows | FCF | Operating cashflows minus capital expenditures | |
| Net cashflows | NCF | Total cash inflows–total cash outflows | |
| Free float shareholders | FFS | Publicly traded common shares | |
| GDP Growth | GDP | % increase (decrease) of GDP from last year | |
| Firm age | AGE | Years from inception | |
| Debt to equity*Net cashflows | DE*NCF | Debt to equity times net cashflows | |

The Error Correction Model

$$\Delta y_t = \gamma_0 + \nu_1 \Delta y_{t-1} + \delta_1 \Delta x_{1,t-1} + \delta_2 \Delta x_{2,t-1} + \ldots \delta_n \Delta x_{n,t-1} + \alpha \varepsilon_{t-1} + \omega_t \tag{2}$$

The adjustment coefficient must be negative. ECM describes how y and x behave in the short-term consistent with a long-run cointegrating relationship. The model of the analysis is written below:

$$\Delta \mathrm{DIV}_t = \gamma_0 + \nu_1 \Delta \mathrm{DIV}_{t-1} + \delta_1 \Delta AT_{1,t-1} + \delta_2 \Delta DE_{2,t-1} + \ldots \delta_n \Delta DTE_{n,t-1} + \alpha \varepsilon_{t-1} + \omega_t$$

The results of the analysis are documented in Table 4.

**4.3.1 Speed of adjustment and Half-life.** The speed of adjustment indicates how fast a system responds to deviations from its normal state while higher speeds indicate faster reversion to equilibrium. Half-life represents the time it takes for half of the deviation to be corrected following a shock. The relationship between the speed of adjustment and half-life is inverse: a faster speed of adjustment directs to a shorter half-life and consequently the variable reverts to equilibrium more quickly. Half-life can be calculated using a specific formula as follows:

$$\text{Half-life} = \frac{\ln(2)}{\ln(\frac{1}{\rho})}, \rho \text{ is speed of adjustment [37]}$$

Related studies on the adoption of half-life and speed of adjustment, for example [38], recommend an empirical framework incorporating the speed of adjustment and the half-life of shocks in stock returns. [37] reveal the concept of permanent and temporary price components relating to the speed of adjustment in financial markets. [39] study the speed of adjustment and half-life and estimate how long stock price takes for firms to close the gap between current and target leverage levels. [40] successfully uses the speed of adjustment in a broader economic model and links it to the half-life of deviations in international stock markets.

For this study, the analysis employs the Fixed Effect Regression Model (FEM) using the ARIMA time series technique, the objective being to solve heteroscedasticity and autocorrelation, which cause an endogeneity problem. Additionally, this study employs the Weighted Least Squares (WLS) technique to confirm that there is no heteroscedasticity in the dataset. The analysis also employs the cointegration concept recommended by [36] to analyze long-term equilibrium relationships. Lastly, the study adopts the concept of speed of adjustment and half-life for the analysis. S2 Fig summarizes all statistical processes that are employed in this study.

The following comprises significant concepts of statistical techniques used in this study.

**4.3.2 Dynamic regression models: ARIMA (Table 3).** Dynamic regression models allow the errors from a regression to contain autocorrelation. To emphasize this change in perspective, an analysis replacing $e_{it}$ with $\omega_t$ is assumed to follow an ARIMA model. $\omega_t$ as an ARIMA (p,d,q) model, we can write

$$Y_{it} = \alpha_i + \beta X_{it} + \delta_t + u_i + \omega_t$$

$$\phi(L)(1 - L)^d \omega_t = \mu + \Theta(L)e_{it}$$

where $e_{it}$ is a white noise series, $\phi(L) = 1 - \phi_1 L - \ldots - \phi_p L^p$, $\Theta(L) = 1 - \theta_1 L - \ldots - \theta_q L^q$

The model has two error terms here—the error from the regression model, denoted by $\omega_t$, and the error from the ARIMA model, denoted by $e_{it}$. Only the ARIMA model errors are assumed to be white noise [41].

**4.3.3 Weighted least squares (Table 3).** Weighted least squares can correct the heteroscedasticity problem with the standard errors. The appropriate estimator is weighted least

squares, which is the implementation of the more general concept of generalized least squares estimator as applied to the least squares model when the covariance matrix of $e_i$ is a positive definite matrix $\Omega$ rather than $\sigma^2 I_N$

$$\hat{\beta}_{GLS} = (X'\Omega^{-1}X)^{-1}(X'\Omega^{-1}Y)$$

The most intuitive approach to GLS is to find the "Cholesky root" matrix $\mathbf{P}$ such that $P'P$ is equal to $\sigma^2\Omega^{-1}$.

This gives $PY = PX\beta + Pe$ for which the OLS estimator is

$$b_{tm} = ((PX)'(PX))^{-1}(PX)'PY$$

$$= (X'P'PX)^{-1}X'P'PY$$
$$== (X'\Omega^{-1}X)^{-1}X'\Omega^{-1}Y = \hat{\beta}_{GLS}$$

In this way the OLS procedure on the transformed model gets the efficient GLS estimator. Thus, WLS consists of OLS on the transformed model:

$$y_i^* = \frac{y_i}{\sigma_i}, x_{i,j}^* = \frac{x_{i,j}}{\sigma_i}, e_i^* = \frac{e_i}{\sigma_i}$$

and $Var(e_i) = \sigma_i^2, Var(e_i^*) = 1$

Thus, dividing each observation by something proportional to the error standard deviation for the observation converts the model to a homoscedastic one [42].

Regarding the heteroscedasticity problem, weighted least squares is an application of the more general concept of generalized least squares.

## 4.4 Data validity and reliability results

First of all, the outlier detection was conducted using the M estimator method, specifically Huber's M-Estimator, with a weighting constant of 1.339. The data in Table 2 were transformed to achieve a normal distribution using the Box-Cox transformation. Afterward, the normality of the data was tested using Zskew and Zkur values. The Zskew or Zkur values should not exceed the range of (-1.96, 1.96) at the significance level of 0.05, or (-2.58, 2.58) at a

**Table 2. Descriptive Statistics and Boxcox transformation (λ).**

| VARIABLES | Boxcox transformation (λ) | | | | | | | | MEAN | SD | MAX | MIN | ZSKEW | ZKUR |
|---|---|---|---|---|---|---|---|---|---|---|---|---|---|---|
| | Agro | Consum | Fin | Indus | Prop | Res | Serv | Techno | | | | | | |
| DIV | 0.93 | 0.86 | 0.79 | 0.82 | 0.88 | 0.87 | 1.02 | 0.87 | 30.49–658.52 | 11.71–33.50 | 147.18 | .13 | -0.36–9.28 | 0.57–20.50 |
| AT | 0.47 | 0.32 | -0.29 | 0.03 | 0.18 | 0.02 | 0.12 | 0.26 | 1.19–0.94 | 0.13–0.32 | 2.76 | .19 | -0.45–0.88 | -2.12–18.2 |
| DE | -0.23 | -0.14 | 0.21 | 0.01 | 0.09 | 0.09 | -0.13 | 0.09 | 0.92–6.64 | 0.10–1.76 | 12.25 | .12 | -0.71–40.33 | -3.49–36.14 |
| FCF | 0.55 | 0.49 | 2.03 | 1.06 | 2.31 | 1.46 | 0.87 | 3.74 | 0.51–2.20 | 0.24–0.95 | 5.13 | .14 | -2.45–10.22 | -2.2–17.17 |
| NCF | 0.86 | 0.77 | 1.12 | 0.61 | 0.28 | 0.33 | 0.49 | 0.93 | 0.68–1.81 | 0.12–0.28 | 2.97 | .01 | 1.57–12.68 | 1.38–61.10 |
| FFS | 0.27 | 0.24 | 0.94 | 0.25 | 0.74 | 0.67 | 0.9 | 0.31 | 2.16–57.15 | 0.12–23.18 | 116.53 | 2.70 | -1.15–0.01 | -3.54–12.5 |
| GDP | 1.14 | 1.03 | 1.08 | 1.04 | 1.12 | 1.08 | 1.03 | 1.1 | 0.97–10.50 | 0.12–5.01 | 15.62 | .18 | -12.98–8.2 | -2.16–40.19 |
| AGE | 1.21 | 1.3 | 0.03 | 0.84 | 0.26 | 0.57 | 0.61 | 1.32 | 2.03–146.65 | 0.28–20.26 | 220.56 | .20 | -7.32–0.94 | -2.51–14.27 |
| DE*NCF | -0.12 | 0.81 | 0.88 | 0.97 | 1.29 | 0.85 | 1.02 | 1.13 | 0.12–2.85 | 0.12–0.59 | 1.40–5.35 | -0.12–0.28 | -0.119–7.36 | -0.247–10.95 |
| No. of OBS. | 374 | 308 | 550 | 814 | 891 | 462 | 1056 | 352 | | | | | | |

significance level of 0.01 [43]. Indicated by the results here is that the dataset is considered a normal distribution as shown in Table 2.

## 5. Findings

As shown in Table 3, the analysis begins with the Hausman test. It emerges that the best method is the Fixed Effect or Random Effect model (FEM) (*p-value <0.05*) [44]. The assumption test is as follows: multicollinearity is a phenomenon characterized by two or more predictors being highly correlated, and if this happens, the standard error of the coefficients will increase; some variables statistically will be insignificant when in fact they should be significant. Linear regression uses tolerances and the variance inflation factor (VIF<10) [45]. Heteroscedasticity refers to the situation where the variance of the error term in a statistical model

**Table 3. Panel regression with ARIMA model–long-run effects.**

| Variables | AGRO | CON | FINANCE | INDUST | PROP | RES | SER | TECH |
|---|---|---|---|---|---|---|---|---|
| Constant | 536.832*** | −32.1970 | 82.3325** | −144.748 | 221.622* | −38.0824 | 100.979* | −6.9411 |
| AT | 3.2761 | **19.565*** | **−2.1755*** | **-66.9154*** | −7.02232 | −20.630 | -1.0831 | 1.821 |
| DE | **−442.172*** | 37.472 | −9.95230 | −73.4429 | **−132.505*** | 76.992 | 63.2345 | 41.433 |
| FCF | 0.3157 | −0.971 | **4.7507 *** | −1.3729 | −0.0033 | −2.4731 | **6.1627** | **0.0466*** |
| NCF | **−22.1819** | -10.413 | -0.0143 | -15.4940 | −53.2120 | **−75.5155*** | 22.064 | 25.281 |
| FFS | −0.0767 | **24.446*** | **0.0986*** | −1.63924 | 0.18582 | 0.004 | −0.151 | −3.699 |
| GDP | **−0.1208*** | **−0.4713*** | **-0.1472*** | **−0.1190*** | **−0.1391*** | **−0.1926*** | **−0.1223*** | **−0.1513*** |
| AGE | **−0.0009** | **-0.0526*** | 42.3675** | 0.01358 | 5.93206* | 0.0455 | 0.007 | 0.015 |
| DE*NCF | **−56.3023** | −15.433 | 0.145939 | −160.475 | 106.419 | **−74.647*** | −6.145 | −24.889 |
| Res_hat | ARIMA (1,1,1) | ARIMA (2,1,0) | ARIMA (1,1,2) | ARIMA (1,1,1) | ARIMA (1,1,1) | ARIMA (1,1,2) | ARIMA (1,1,1) | ARIMA (1,1,1) |
| DIV$_{(t-1)}$ | 0.7319*** | 0.6654*** | 0.8428*** | 0.6945*** | 0.5320*** | 0.7098*** | 0.7651*** | 0.6906*** |
| DIV$_{(t-2)}$ | | 0.1776** | | | | | | |
| $\varepsilon_{t-1}$ | −0.2189* | -0.3215*** | −.2430*** | −0.2910*** | −0.3650*** | −0.3143*** | −0.2463*** | -0.3261** |
| $\varepsilon_{t-2}$ | | | 0.2430*** | | | 0.1144*** | | |
| Fixed effects Hausman test (*p-value*) | (0.0022) | (0.0041) | (0.0305) | (0.0001) | (0.0003) | (0.0000) | (0.0002) | (0.0010) |
| R$^2$ | 0.7437 | 0.5716 | 0.4929 | 0.5170 | 0.5871 | 0.4690 | 0.4849 | 0.6638 |
| Adj R$^2$ | 0.7374 | 0.5540 | 0.4823 | 0.5085 | 0.5824 | 0.4536 | 0.4801 | 0.6549 |
| Durbin-Watson | 2.0310 | 2.2640 | 2.2290 | 2.0100 | 2.0710 | 2.1110 | 2.0620 | 2.0880 |
| Autocorrelation | Ljung-Box Q' = 4.833 (0.4365) | Ljung-Box Q' = 3.476 (0.9424) | Ljung-Box Q' = 8.419 (0.4925) | Ljung-Box Q' = 6.268 (0.2810) | Ljung-Box Q' = 5.811 (0.4446) | Ljung-Box Q' = 8.946 (0.1766) | Ljung-Box Q' = 11.704 (0.1649) | Ljung-Box Q' = 3.636 (0.6028) |
| ADF TEST STATIONARY | tau c (1) = -19.242 (0.000) | tau_c (1) = -17.445 (0.000) | tau_c (1) = -23.286 (0.000) | tau_c (1) = -28.501 (0.000) | tau_c (1) = -30.420 (0.000) | tau_c (1) = -21.980 (0.000) | tau_c (1) = -31.997 (0.000) | tau_c (1) = -18.661 (0.000) |
| RMSE | 14.658 | 11.624 | 7.597 | 8.99 | 12.013 | 17.524 | 10.235 | 11.714 |
| AIC | 3111.9417 | 2392.3804 | 3796.2899 | 5927.2962 | 7041.7412 | 3637.6825 | 2633.5995 | 2729.7318 |
| BIC | 3123.7144 | 2440.8294 | 3852.2952 | 5979.0042 | 7094.4446 | 3687.2833 | 2688.1738 | 2776.0612 |
| HQC | 3116.6160 | 411.7547 | 3818.1776 | 5947.1449 | 7061.8851 | 3657.2125 | 2654.2875 | 2748.1706 |

FE: Fixed Effect Model

* significance at the 0.05 level

** significance at the 0.01 level

*** significance at the 0.001 level

is not constant across all values of the predictor variables. This can lead to inaccurate results and invalid conclusions in the model. In linear regression using the Breusch-Pagan Test (p-value<0.05), the null hypothesis residual is constant variance and in the panel using the Wald test (p-value<0.05).

Autocorrelation refers to correlation between the values of the same variables across different observations in the data. An autocorrelation problem that is known as positive autocorrelation leads to underestimating the standard error of the mean; alternatively, serial correlation causes the standard errors of the coefficients to be smaller than they actually are and higher R-squared. Linear regression uses Durbin-Watson (1.5–2.5) [45] and [46] for autocorrelation in panel data (*p-value >0.05*). Pesaran CD (cross-sectional dependence) tests serve to test whether the residuals are correlated across entities. Cross-sectional dependence can lead to bias in tests results (also called contemporaneous correlation). The null hypothesis is that residuals are not correlated (p-value>0.05) [47].

Autocorrelation problem uses the residual for the ARIMA model in time series: AGRO has ARIMA (1,1,1), CON has ARIMA (2,1,0), FINANCE has ARIMA (1,1,2), INDUST has ARIMA (1,1,1), PROP has ARIMA (1,1,1), RES has ARIMA (1,1,2), SER has ARIMA (1,1,1) and TECH has ARIMA (1,1,1). The models selected via Akaike Information Criterion (AIC), Bayesian Information Criteria (BIC) and Hannan-Quinn Criterion (HQC) are the smallest. The P-value of Ljung-Box Q' is >0.05, and therefore no autocorrelation in all models is found. ADF TEST (p-value<0.05) means that the residual of models is stationary. Heteroscedasticity when corrected by weighted least squares (WLS) means that the weight varies with the residual as shown in Table 3.

## 5.1 Regression long-run analysis

Table 3 summarizes the research results using a panel regression with the ARIMA model and presents a detailed analysis across various sectors: Agriculture (AGRO), Consumer (CON), Finance (FINANCE), Industrial (INDUST), Property (PROP), Resources (RES), Services (SER), and Technology (TECH). Each sector exhibited distinct responses to the examined variables, highlighting sector-specific influences on dividend payout policy.

Asset turnover (AT) exerts a favorable and significant influence on the dividend distribution policy within the CON sector. Indicated here is that firms in this industry derive advantages from asset usage. Conversely, in the FIN and INDUST sectors, AT adversely impacts dividend payment policy, resulting in a reduction in dividend payouts. The impact of debt to equity (DE) varies across different sectors. The AGRO industry witnessed a substantial adverse impact on its dividend distribution strategy. In the agriculture industry, increased indebtedness diminishes dividend payouts. Conversely, the PROP sector had a positive correlation between DE and the dividend distribution policy. Shown here is that PROP companies are more likely to utilize debt more efficiently to enhance dividend payout policy. Free cashflows (FCF) positively and significantly relate to the dividend payout policy in the FINANCE, SER, and TECH sectors. This suggests that the excess cash flows after the payment of capital expenditures strengthen the dividend payout policy. Ultimately, net cash flows (NCF) had a varied outcome. In the AGRO and RES sectors, there exists a markedly unfavorable correlation concerning the dividend distribution policy. This strongly suggests that net cash flows negatively influence the dividend distribution policy in AGRO and RES companies.

The effect of free float shareholders (FFS) is especially significant in the CON and FINANCE sectors, where it favorably affects dividend payment policy. It means that an increased proportion of shares accessible for trading enhances dividend payouts in these industries. Furthermore, GDP growth demonstrated a consistently adverse impact across all

sectors. The persistent negative correlation regarding dividend payout policy suggests that economic development or expansion negatively influences dividend payout policy across all industries, underscoring the susceptibility of dividend payout policy to overarching macroeconomic conditions. Firm age (AGE) significantly relates to the dividend payout policy in a positive way in the FINANCE and PROF sectors. Conversely, Firm age negatively affects dividend payout policy in the ARGO and CON sectors. It suggests that younger companies are more likely to operate their businesses in such a way so that a dividend payout policy is suitable. Lastly, the interaction between debt to equity and net cashflows (DTE) impacts on dividend payment policy in a negative manner in the AGRO and RES sectors. This highlights the potential pressure that leverage and cashflows limitations exert.

## 5.2 Short-run and speed of adjustment to half-life

Table 4 shows short-run and speed of adjustment to long-run equilibrium. For the short-run adjustment of AGRO, when no other factors are involved, DIV has a rate of change of 510.3670 units. For a 1 unit change in DE, DIV will decrease 4.68 times. For a 1 unit change of dividend payout (t-1), this will increase DIV by 0.6840 units. DE affects the adjustment of DIV with a rate of adjustment to short-run equilibrium of 0.12% (Half-life = 0.1031 year) (or

**Table 4. Short-run and speed of adjustment to long-run equilibrium.**

| Variables | AGRO | CON | FINANCE | INDUST | PROP | RES | SER | TECH |
|---|---|---|---|---|---|---|---|---|
| const | 510.3670*** (193.4330) | −21.6302 (34.1645) | −2.0362 (29.0777) | 69.2647 (200.3910) | 137.9800* (62.3398) | −110.8940 (83.1292) | 2.1930 (46.5222) | −5.8929 (71.6935) |
| ΔAT | 0.9654 (3.1673) | 15.0936* (7.1150) | −1.5358 (1.1419) | 17.5042 (28.1244) | −2.1716 (3.7844) | −22.2285 (27.1441) | 9.7586 (7.8864) | −0.6072 (2.6269) |
| ΔDE | −468.250** (145.273) -0.0012† | 9.0648 (27.2249) | 15.5629 (12.4911) | −65.7631** (26.3030) -0.0014† | −136.999* (62.5333) -0.0049† | -142.193* (77.7299) -0.0091† | 20.3225 (47.8290) | 20.0159 (68.2083) |
| ΔFCF | 5.6196 (3.6823) | 0.2952 (4.4222) | 1.8653** (0.6620) -0.0116† | −1.15867 (1.6003) | −0.0964 (0.1848) | −0.6941 (0.9872) | 6.2183 (2.3217) | 0.0574*** (0.0154) -0.0207† |
| ΔNFC | −10.4345 (12.9450) | −12.2938 (22.5756) | 8.2685 (8.0049) | 54.3588 (241.8370) | −71.4643 (64.3161) | 138.942* (65.6063) | 1.1080 (28.3136) | 14.2205 (33.3254) |
| ΔFFS | 0.0332 (0.0840) | 9.4341** (3.3707) | 0.0263 (0.0201) | 0.3285 (1.3309) | 0.0354 (0.0754) | −0.0178 (0.1737) | −0.0344 (0.0566) | −1.4217 (1.5696) |
| ΔGDP | 0.0086 (0.2749) | −0.3335* (0.1577) | 0.0828 (0.0639) | −0.1538* (0.0696) | −0.1457* (0.0710) | −0.1457 (0.1132) | −0.2135 (0.1601) | −0.2363* (0.1136) |
| ΔAGE | −0.0001 (0.0007) | −0.0209* (0.0108) -0.2116† | −4.4460 (20.5973) | −0.0499* (0.0208) -0.1943† | 2.4085 (1.5287) | 0.4451 (0.3171) | −0.1088 (0.2626) | −0.0017 (0.0106) |
| ΔDE*NCF | −21.0604 (47.5078) | 4.3972 (20.7013) | −9.0460 (5.9817) | −63.6724 (242.6000) | 87.8042 (64.9440) | −132.199* (62.9502) | −9.0728 (29.9946) | −12.2424 (31.7252) |
| ΔDIV$_{(t-1)}$ | 0.6840*** (0.0375) -0.2371† | 0.6087*** (0.0567) -0.1611† | 0.7041*** (0.0293) -0.1040† | 0.6767*** (0.0252) -0.1121† | 0.5542*** (0.0274) -0.1533† | 0.6145*** (0.0369) -0.1472† | 0.7081*** (0.0217) -0.1321† | 0.6652*** (0.0389) -0.1811† |
| $\varepsilon_{t-1}$ | −0.1256*** (0.0223) | -0.1431*** (0.0637) | −0.0765*** (0.0339) | −0.1550*** (0.0228) | −0.1220*** (0.0422) | −0.1134*** (0.0409) | −0.1413*** (0.0512) | -0.1927*** (0.0219) |
| Adj R² | 0.6008 | 0.5876 | 0.5764 | 0.4998 | 0.5310 | 0.4223 | 0.5221 | 0.5056 |
| Durbin-Watson | 1.9431 | 1.9561 | 2.2362 | 2.0014 | 2.0388 | 2.0505 | 2.0554 | 2.0754 |

* Significance at the 0.05 level

*** significance at the 0.001 level, standard error in the parentheses

† is the speed of adjustment.

adjusts to equilibrium by 50% of the deviation from equilibrium) and dividend payout (t-1) affects the adjustment of DIV with a rate of adjustment to short-run equilibrium of 23.71% (Half-life = 0.4816 year). If any event or effect causes the relationship to deviate from the original equilibrium, DIV ($\varepsilon_{t-1}$) will adjust back to long-run equilibrium with a rate of 12.56% (Half-life = 0.3341 year).

For the short-run adjustment of the consumer sector, a change of AT in 1 unit will increase DIV by 15.0936 units, while if there is a change of FFS in 1 unit, DIV will increase DIV by 9.4341 units. A change of GDP in 1 unit, DIV will decrease by 0.3335 units. In addition, a change of AGE in 1 unit means that DIV will decrease DIV by 0.0209 units. A change of dividend payout (t-1) increases 1 unit, DIV will increase by 0.6087 units. AGE affects the adjustment of DIV with a speed of adjustment to short-run equilibrium of 21.16% (Half-life = 0.4463 year) (or adjusts to equilibrium by 50% of the deviation from equilibrium) and dividend payout (t-1) affects the adjustment of DIV with a speed of adjustment to short-run equilibrium of 16.11% (Half-life = 0.3797 year). If any event or shock causes the relationship to deviate from the original equilibrium, the value of DIV ($\varepsilon_{t-1}$) rebounds back to long-term equilibrium at a rate of 14.31% (Half-life = 0.3565 year).

Referring to the short-run adjustment of the finance sector, for a 1-unit change in FCF, DIV increases by 1.8653 units, a 1-unit change in dividend payout (t-1) increases by 0.7041 units. FCF affects the adjustment of DIV at a rate of 1.16% (Half-life = 0.1555 year) (or 50% of the deviation from equilibrium) and dividend payout (t-1) affects the adjustment of DIV at a rate of 10.40% (Half-life = 0.3062 year). If any event or shock causes the relationship to deviate from the original equilibrium, the value of DIV ($\varepsilon_{t-1}$) adjusts back to the long-run equilibrium at a rate of 7.65% (Half-life = 0.2697 year).

For the short-run adjustment of the industrial sector, a 1 unit change in DE will decrease DIV by .65 times, while a 1 unit change in GDP will diminish DIV by 0.1538 units. A 1 unit change in AGE will diminish DIV by 0.0499 units, and an increase in dividend payout (t-1) by 1 unit will increase DIV by 0.6767 units. DE affects the adjustment of DIV by the speed of its short-run equilibrium adjustment of 0.14% (Half-life = 0.1055 year) (or adjusts to equilibrium by 50% of the deviation from equilibrium). AGE affects the adjustment of DIV by the speed of its short-run equilibrium adjustment of 19.43% (Half-life = 0.4231 year) (or adjusts to equilibrium by 50% of the deviation from equilibrium), and dividend payout (t-1) affects the adjustment of DIV by the speed of its short-run equilibrium adjustment of 11.21% (Half-life = 0.3167 year). If any event or shock occurs that causes the relationship to deviate from the original equilibrium, DIV($\varepsilon_{t-1}$) will adjust itself back to the long-run equilibrium at a speed of 15.50% (Half-life = 0.3718 year).

With reference to the short-run adjustment of the property sector, when no other factors are involved, DIV exhibits a rate of change of 137.9800 units. For a 1 unit change in DE, DIV decreases by 1.36 times, a 1 unit change in GDP decreases DIV by 0.1457 units, and a 1 unit increase in dividend payout (t-1) will increase DIV by 0.5542 units. DE affects the adjustment of DIV with a speed of 0.49% (Half-life = 0.1303 year) (or 50% of the deviation from equilibrium), and $y_{t-1}$ affects the adjustment of DIV with a speed of 15.33% (Half-life = 0.3696 year). If any event or shock causes the relationship to deviate from the original equilibrium, the value of DIV ($\varepsilon_{t-1}$) adjusts back to the long-run equilibrium with a speed of 12.20% (Half-life = 0.3295 year).

Concerning the short-run adjustment of the resource sector, for a 1 unit change in DE, DIV will decrease by 1.42 times, a 1 unit change in NCF will increase DIV by 138.942 units, a 1 unit change in DE*NFC will decrease DIV by 132.199 units, and the change of dividend payout (t-1) will increase in DIV by 0.6145 units. DE affects the adjustment of DIV with the short-run equilibrium adjustment rate of 0.91% (Half-life = 0.1475 year) (or 50% of the deviation

from equilibrium), and dividend payout (t-1) affects the adjustment of DIV with a short-run equilibrium adjustment rate of 14.72% (Half-life = 0.3618 year). If any shock causes the relationship to deviate from the original equilibrium, the value of DIV ($\varepsilon_{t-1}$) will adjust back to the long-run equilibrium with a speed of 11.34% (Half-life = 0.3184 year).

For the short-run adjustment of the service sector, a 1 unit increase in dividend payout (t-1) will increase DIV by 0.7081 units, dividend payout (t-1) affects the adjustment of DIV with the short-run equilibrium adjustment rate of 13.21% (Half-life = 0.3424 year). If any event or shock causes the relationship to deviate from the original equilibrium, the DIV ($\varepsilon_{t-1}$) will adjust back towards the long-run equilibrium with a rate of 14.13% (Half-life = 0.3542 year).

Referring to the short-run adjustment of the technology sector, for a 1-unit change in FCF, DIV increases by 0.0574 units, a 1-unit change in GDP decreases DIV by 0.2363 units, and a 1-unit change in dividend payout (t-1) will increase DIV by 0.6652 units. FCF affects the adjustment of DIV with a short-run equilibrium adjustment speed of 2.07% (Half-life = 0.1788 year) (or adjusts to equilibrium by 50% of the deviation from equilibrium) and dividend payout (t-1) affects the adjustment of DIV with a short-run equilibrium adjustment speed of 18.11% (Half-life = 0.4057 year). If there is any event or shock that causes the relationship to deviate from the original equilibrium, the value of DIV($\varepsilon_{t-1}$) adjusts back to the long-run equilibrium with a speed of 19.27% (Half-life = 0.4210 year).

## 6. Discussion

### 6.1 Long-term perspective

Table 4 illustrates that GDP growth has a correlation with dividend distribution policy in a positive way and in accordance with the long-term view. This finding is in line with [11] and it means long-term GDP growth is more likely to adversely affect dividend payout policies. This is because companies would set as their priority the need to reinvest their gains in new projects during times of economic growth instead of distributing dividends. Moreover, investors in high-growth periods are more likely to choose long-term capital gains instead of immediate dividends. This causes companies to reduce their dividend payouts and instead concentrate on reinvestment.

In the event of a GDP decline, companies may be more likely to provide elevated dividends to convey stability to investors, assuring them of the firm's financial well-being despite the economic recession. During such periods, companies frequently encounter a scarcity of lucrative investment prospects, prompting them to distribute surplus income to shareholders instead of reinvesting it. Shareholders may also request increased dividends as recompense for diminished capital gains, particularly during periods of market volatility. Furthermore, firms with consistent cash flows may augment dividends to underscore their robustness. Companies that emphasize sustaining or augmenting dividends, especially throughout economic downturns, do so to: firstly, fulfill market expectations; and secondly, attract income-oriented investors, so rendering dividend-paying equities more attractive during periods of economic volatility.

Alongside establishing GDP development as a long-term plan, dividend payment policy is shaped by certain sectors of the economy as follows. The asset turnover ratio has a favorable correlation with dividend payout in industries such as retail, where effective asset utilization results in increased earnings and greater cash availability for dividends. In capital-intensive sectors such as the property sector, higher asset turnover potentially results in a necessity for reinvestment, consequently diminishing dividend payout. In utilities and property sectors, an elevated debt-to-equity ratio may correlate favorably with dividend payouts because companies use debt for expansions, while cashflows are preserved for dividend payouts. On the other hand, in sectors with unstable profitability such as the property market, a raised debt-to-equity

ratio potentially causes financial burden. This results in diminished dividend payout. Industries with stable and mature status such as consumer products are more likely to have positive free cashflows resulting in increased dividend payout. Conversely, businesses emphasizing growth and reinvestment, such as biotechnology or technology, may emphasize internal investments, leading to diminished dividends despite negative free cash flows.

Favorable net cashflows in established industries such as utilities make increased dividends possible, since corporations own greater liquid assets for distribution. In growth-oriented industries such as technology, negative net cash flows resulting from substantial investment may lead to diminished or nonexistent dividend distributions. Sectors characterized by a substantial proportion of free float shareholders, such as consumer products, may experience heightened pressure to sustain or augment dividends to entice investors. In sectors such as family-owned enterprises, less free float shareholders may reduce the emphasis on dividends because major shareholders are more likely to favor reinvestment or maintaining control. Companies in mature industries, for instance the manufacturing sector, are more likely to demonstrate consistent earnings and elevated dividend payout. Emerging companies in dynamic industries like technology are more likely to reinvest profits for the purposes of expansion.

## 6.2 Short-term perspective

In the short-run, the study finds that past dividend payments significantly influence dividend payment policy. This aligns with the previous study by [30]. Additionally, debt-to-equity ratio positively relates to dividend payout policy in the industrial, property, and resource sectors. This indicates that the companies potentially employ debt as operations and expansion, protecting capital for dividends. Free cashflows are positively associated with dividend payout policies in the finance and technology industries. It means that companies with extensive free cashflows retain excess cash after paying operational expenses and capital expenditures and result in dividend distributions. Lastly, aged companies positively relate to dividend payout policies in the consumer and industrial sectors. In essence, aged firms exhibit continuing ability to make profits and it means that they are able to maintain or increase their dividend distributions.

## 6.3 Contributions

**6.3.1 Theoretical contributions.**   This research supports the dividend signaling theory indicating that GDP growth, firm age, financial ratios, and free float shareholders significantly relate to dividend payout policy. For instance, GDP growth is more likely to affect dividends. In addition, older firms and financial metrics such as asset turnover anda cashflows are associated with dividend payout policy. Additionally, free float shareholders are related to dividends payout policy. This research proves the value relevance of accounting information indicating that financial ratios are significant foundations of dividend payout policy. Consequently, this study emphasizes the value relevance of accounting information and especially with reference to dividend payout policy.

**6.3.2 Recommendations for practical implications and unique contributions.**   *For corporate managers in Emerging economies*. This study offers management teams practical information to match dividend distribution plans with sectoral and macroeconomic circumstances. When considering dividend policy, management teams should closely watch GDP patterns as a strategic guide. Strong economic situations are more likely to care more about keeping more profits for the purposes of reinvestment. On the other hand, keeping or raising dividends indicates financial stability and means that income-oriented investors are subjected to economic

instability. Furthermore, for sector-specific approaches, enhancing asset utilization can support greater dividends while ensuring operational efficiency in sectors with high asset turnover (e.g., consumer sector). In addition, leveraging excess liquidity can aid in strengthening investor confidence in sectors with stable free cashflows (e.g., finance and services). Stable dividend payouts should be given top priority for mature companies in the property and industrial sectors as these reflect shareholder expectations in reputable companies. In sectors with large free float investors, manager teams should also exercise caution because continuous dividend distributions could be considered as strong financial planning. Also, long-term financial sustainability depends on balancing operational and macroeconomic considerations with the needs of several stakeholders.

*For policymakers in Emerging economies.* The findings of this study might help emerging nations' regulatory authorities create and implement policies that lead to robust and healthy financial markets. Policymakers should motivate businesses to use dividend policies appropriate for economic cycles, therefore promoting financial stability across both economic ups and downs. This can involve setting rules for companies to balance reinvestment needs with shareholder returns during times of GDP growth, giving incentives for sectors that give sustainable dividend policies top priority. Especially, the policymakers should monitor in sectors vital to economic stability with contemporary research on sector-specific effects of macroeconomic conditions on dividend policies to improve regulatory frameworks. By combining regulatory programs with economic and sectoral underlying forces, policymakers can lower risks related with essential dividend programs and encourage investor confidence in emerging economies.

**6.3.3 Articulating unique contributions.** This work provides several novel contributions to the corpus of present studies on dividend paying policies.

1. This study separates the investigation by sector, revealing intricate relationships between dividend distribution regulations and macroeconomic variables unlike other studies extending conclusions across sectors. For instance, the different consequences of debt-to-equity ratios and asset turnover in different industries highlight the need of sector-specific strategies.

2. By stressing the creation and market-driven economies of this article, which thus offers insights distinct from those in developed markets, it answers a major gap in the research. It reveals that GDP growth affects dividend policy in various ways.

3. ARIMA and Weighted Least Squares (WLS) solve limitations in traditional regression models including autocorrelation and heteroscedasticity. These techniques result in more reliable outcomes.

4. Applying the dividend signaling and the value relevance of accounting information theories empirically verifies how dividend policy interacts with free cashflows, GDP growth, and free float shareholders.

This study significantly contributes to existing knowledge. It not only increases the theoretical comprehension of dividend payout policy but also provides management teams, policymakers and stakeholders with the means to shift the demanding dynamics of growing capital markets in useful ways.

## 6.4 Conclusions

In the long-term, companies frequently reduce dividend payments during periods of GDP growth in order to prioritize reinvestment in expansion-related projects rather than distributing dividends. Asset turnover positively impacts dividend payouts in sectors such as retail

resulting in increased profits. However, in capital-intensive industries, it may result in reduced payouts due to reinvestment requirements. In sectors such as utilities, where debt is used to finance development, a higher debt-to-equity ratio can facilitate dividend distributions. However, in volatile sectors like technology, it may result in a decline in payouts. Dividends are more likely to increase due to optimistic free cashflows and net cashflows in the manufacturing sector, while industries with more potential growth are more likely to set reinvestment as their priority rather than pay dividends. In the consumer sector, companies are potentially under pressure to sustain dividends from high-free float shareholders. Finally, younger firms in dynamic sectors reinvest profits resulting in reduced dividends, while on the other hand aged firms in mature sectors of the economy typically have larger dividend payouts.

In the short-term, past dividend payments significantly influence dividend payment polily. Additionally, companies with high debt-to-equity ratio potentially facilitate dividend disbursements by allowing companies to manage debt for operations and growth, thereby preserving cash for dividends. Dividends are more likely to be maintained or increased by older firms due to their established operations and having a history of stable earnings. In the same way, increased free cashflows generate surplus liquidity, which enables organizations to distribute dividends without experiencing financial strain.

## 6.5 Limitation of this study and further studies

The use of archival information in this study presents certain restrictions. This may suffer from outdated data resulting in the findings not being completely reliable. The nature of the information may not clearly depict the causality between the independent and dependent variables. Moreover, variables may not capture all relevant factors and change over time, thus causing possible measurement error or omitted variable bias. Further studies should adopt more recent measures such as corporate governance indicators and ESG, lag dividend payments. Lastly, the statistical methods used may be appropriate in different datasets. Assumptions test need to be conducted carefully. For example, GMM is also powerful. This approach is well-suited for capturing the primary dynamics relevant to our analysis. Recognizing these steps should offer a more comprehensive perspective on the subject that has been investigated in this paper.

## Supporting information

**S1 Fig. Conceptual framework.**
(TIF)

**S2 Fig. Flows of statistical techniques adopted in this study.**
(TIF)

**S1 Data.**
(XLSX)

## Acknowledgments

The authors thank Dr. Marco Realdon, Brunel University London and the Faculty of Liberal Art, Rajamangala University of Technology Rattanakosin for their time and opportunity in helping with this paper.

## Author Contributions

**Conceptualization:** Kittisak Jangphanish, Wachira Boonyanet, Supa Tongkong.

**Data curation:** Kittisak Jangphanish.

**Formal analysis:** Kittisak Jangphanish, Wachira Boonyanet, Supa Tongkong.

**Methodology:** Kittisak Jangphanish.

**Project administration:** Wachira Boonyanet.

**Software:** Kittisak Jangphanish.

**Validation:** Wachira Boonyanet, Supa Tongkong.

**Writing – original draft:** Kittisak Jangphanish, Wachira Boonyanet.

**Writing – review & editing:** Wachira Boonyanet, Supa Tongkong.

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
