## [Decision Letter · Decision Letter 0]

12 Nov 2024

PONE-D-24-42377Analyzing Robust Dividend Payout Policy with Dynamic Panel Regression Application of Speed of Adjustment to Half-LifePLOS ONE

Dear Dr. Boonyanet,

Thank you for submitting your manuscript to PLOS ONE. After careful consideration, we feel that it has merit but does not fully meet PLOS ONE’s publication criteria as it currently stands. Therefore, we invite you to submit a revised version of the manuscript that addresses the points raised during the review process.

We look forward to receiving your revised manuscript.

Kind regards,

Michel Alexandre, Ph.D

Academic Editor

PLOS ONE

Journal Requirements:

2. In the online submission form, you indicated that your data is available only on request from a third party. Please note that your Data Availability Statement is currently missing the contact details for the third party, such as an email address or a link to where data requests can be made. Please update your statement with the missing information.

Reviewers' comments:

Reviewer's Responses to Questions

**Comments to the Author**

1. Is the manuscript technically sound, and do the data support the conclusions?

Reviewer #1: Yes

Reviewer #2: No

2. Has the statistical analysis been performed appropriately and rigorously? 

Reviewer #1: Yes

Reviewer #2: No

3. Have the authors made all data underlying the findings in their manuscript fully available?

Reviewer #1: Yes

Reviewer #2: Yes

4. Is the manuscript presented in an intelligible fashion and written in standard English?

Reviewer #1: Yes

Reviewer #2: No

5. Review Comments to the Author

Reviewer #1: 1. Introduction: The introduction is well outlined. However, "Problem Statement" and "Objective" of the article need to be clearly defined.

2. Literature Review: The review of empirical research is well explained with critical justification. Underlying theories were also explained. However, a clear literature gap is still shallow in the description. The author is advised to clearly identify and outline the literature gap.

3. Conceptual Framework: The conceptual Framework is developed. In addition, the author is advised to include the hypotheses.

4. Research Methodology: Excellent and rigorous statistical analysis techniques were followed.

5. Findings: Reasonably well written technical analysis on findings. No further discussion is required.

6. Discussion: Excellent discussions on theoretical and practical contributions, limitations and scope for further research. No edit is required.

Reviewer #2: Review

Dear authors,

This research article provides a well-rounded background on the complexities and importance of dividend payout policy in the context of corporate finance. The study is commendable for recognizing both financial and operational factors, such as asset turnover, firm age, and GDP growth, which are often overlooked in the dividend payout policy literature. Additionally, the introduction of advanced statistical techniques like ARIMA, Weighted Least Squares, and the use of instrumental variables for handling endogeneity reflects a sophisticated approach. By targeting emerging economies and using a variety of variables, the research fills notable gaps in our understanding of dividend policy in diverse economic contexts. However, a few critiques may be considered:

1. Although ARIMA, WLS, and IV techniques are robust for addressing endogeneity, relying on ARIMA and WLS models might not sufficiently capture all nuances in panel data, especially when dealing with cross-sectional variations. More advanced econometric techniques, like dynamic panel GMM, might offer further insights in handling panel data complexities.

2. The study mentions the dividend signaling theory and accounting information relevance theory; however, it lacks a deeper discussion of how these theories interact with the empirical model. A more elaborate theoretical framework could provide clearer motivation for selecting specific variables.

3. While the paper provides a solid academic perspective, it would benefit from a stronger focus on the implications for corporate managers and policymakers in emerging economies, beyond general recommendations.

4. The study indeed falls short in articulating its unique contributions to the existing body of research on dividend payout policy. Without a clear discussion on how it advances or diverges from prior studies, the study may struggle to demonstrate its significance within the field.

Overall, while the research is thorough, it could be further enriched by expanding the theoretical foundation, refining the methodology, and enhancing practical applications.

6. PLOS authors have the option to publish the peer review history of their article (what does this mean?). If published, this will include your full peer review and any attached files.

Reviewer #1: No

Reviewer #2: No

---

## [Author Response · Author response to Decision Letter 0]

25 Nov 2024

Response to reviewers

Reviewer #1: 

1. Introduction: The introduction is well outlined. However, "Problem Statement" and "Objective" of the article need to be clearly defined.

Revised as recommended. (pages 2 and 3)

2. Literature Review: The review of empirical research is well explained with critical justification. Underlying theories were also explained. However, a clear literature gap is still shallow in the description. The author is advised to clearly identify and outline the literature gap.

Revised as recommended. (pages 6 and 7)

3. Conceptual Framework: The conceptual Framework is developed. In addition, the author is advised to include the hypotheses.

Thank you for the suggestion to include hypotheses in the manuscript. While we appreciate the importance of clearly stating hypotheses in many research contexts, this study is structured around a data-driven exploration of financial information and previously successful factors. This approach allows for a broader examination of these relationships, beyond a confirmatory framework, and is consistent with our research objectives. Therefore, we have opted to focus on the empirical findings and interpretations without predefined hypotheses to maintain the exploratory nature of our study.

4. Research Methodology: Excellent and rigorous statistical analysis techniques were followed.

Thank you.

5. Findings: Reasonably well written technical analysis on findings. No further discussion is required.

Thank you.

6. Discussion: Excellent discussions on theoretical and practical contributions, limitations and scope for further research. No edit is required.

Thank you.

Review#2

Dear authors,

This research article provides a well-rounded background on the complexities and importance of dividend payout policy in the context of corporate finance. The study is commendable for recognizing both financial and operational factors, such as asset turnover, firm age, and GDP growth, which are often overlooked in the dividend payout policy literature. Additionally, the introduction of advanced statistical techniques like ARIMA, Weighted Least Squares, and the use of instrumental variables for handling endogeneity reflects a sophisticated approach. By targeting emerging economies and using a variety of variables, the research fills notable gaps in our understanding of dividend policy in diverse economic contexts. However, a few critiques may be considered:

1. Although ARIMA, WLS, and IV techniques are robust for addressing endogeneity, relying on ARIMA and WLS models might not sufficiently capture all nuances in panel data, especially when dealing with cross-sectional variations. More advanced econometric techniques, like dynamic panel GMM, might offer further insights in handling panel data complexities.

Thank you for your insightful comments regarding our methodological approach. We recognize that dynamic panel GMM offers robust tools for addressing certain aspects of panel data, especially with cross-sectional variations and endogeneity. However, in this study, we selected ARIMA, WLS, and IV techniques to address endogeneity and autocorrelation, with an emphasis on model simplicity and interpretability in line with our specific data structure and research objectives. 

Panel data, which combines cross-sectional and time series dimensions, is often associated with endogeneity issues. Endogeneity can lead to challenges such as autocorrelation and heteroscedasticity. While we acknowledge that GMM is a robust method to address endogeneity, it is important to note that these problems can also be effectively managed through alternative techniques. For instance, autocorrelation can be addressed using methods such as Generalized Least Squares, Exponential Smoothing, Spectral Analysis, Fourier Analysis, or ARIMA (Wiboonpongse et al., 2015; Barros et al., 2015). Similarly, heteroscedasticity can be mitigated with approaches like Weighted Least Squares (WLS), Heteroscedasticity-Consistent Standard Errors (HGL), and Driscoll-Kraay standard errors (Gill, 2007; Hoechle, 2007; Wooldridge, 2007). These techniques aim to ensure that estimators possess desirable properties, particularly being BLUE (Best Linear Unbiased Estimator).

While GMM is also powerful, we believe that our approach is well-suited for capturing the primary dynamics relevant to our analysis. Nonetheless, we mention this suggestion in future work to further enhance robustness. (page 25)

Refrrences:

Barros, L. A., Bergmann, D. R., Castro, F. H., & Silveira, A. D. M. D. (2020). Endogeneity in panel data regressions: methodological guidance for corporate finance researchers. Revista brasileira de gestão de negócios, 22, 437-461.

Hoechle, D. (2007). Robust Standard Errors for Panel Regressions with Cross-Sectional Dependence. The Stata Journal, 7, 281-312

Gill, J., & Leemann, L. (2001). Weighted Least Squares. In Generalized Linear Models: A Unified Approach (pp. 42-44). SAGE, Thousand Oaks, California.

Wiboonpongse, A., Liu, J., Sriboonchitta, S., & Denoeux, T. (2015). Modeling dependence between error components of the stochastic frontier model using copula: Application to intercrop coffee production in Northern Thailand. International Journal of Approximate Reasoning, 65, 34-44.

Wooldridge, J. M. (2010). Econometric analysis of cross section and panel data. MIT Press, Cambridge.

2. The study mentions the dividend signaling theory and accounting information relevance theory; however, it lacks a deeper discussion of how these theories interact with the empirical model. A more elaborate theoretical framework could provide clearer motivation for selecting specific variables.

Revised as recommended (pages 4 and 5).

3. While the paper provides a solid academic perspective, it would benefit from a stronger focus on the implications for corporate managers and policymakers in emerging economies, beyond general recommendations.

Revised as recommended (page 22).

4. The study indeed falls short in articulating its unique contributions to the existing body of research on dividend payout policy. Without a clear discussion on how it advances or diverges from prior studies, the study may struggle to demonstrate its significance within the field.

Revised as recommended (page 23).

---

## [Decision Letter · Decision Letter 1]

12 Dec 2024

Analyzing Robust Dividend Payout Policy with Dynamic Panel Regression Application of Speed of Adjustment to Half-Life

PONE-D-24-42377R1

Dear Dr. Boonyanet,

We’re pleased to inform you that your manuscript has been judged scientifically suitable for publication and will be formally accepted for publication once it meets all outstanding technical requirements.

Kind regards,

Michel Alexandre, Ph.D

Academic Editor

PLOS ONE

Reviewers' comments:

Reviewer's Responses to Questions

**Comments to the Author**

1. If the authors have adequately addressed your comments raised in a previous round of review and you feel that this manuscript is now acceptable for publication, you may indicate that here to bypass the “Comments to the Author” section, enter your conflict of interest statement in the “Confidential to Editor” section, and submit your "Accept" recommendation.

Reviewer #1: All comments have been addressed

Reviewer #2: All comments have been addressed

2. Is the manuscript technically sound, and do the data support the conclusions?

Reviewer #1: Yes

Reviewer #2: Yes

3. Has the statistical analysis been performed appropriately and rigorously? 

Reviewer #1: Yes

Reviewer #2: Yes

4. Have the authors made all data underlying the findings in their manuscript fully available?

Reviewer #1: Yes

Reviewer #2: Yes

5. Is the manuscript presented in an intelligible fashion and written in standard English?

Reviewer #1: Yes

Reviewer #2: Yes

6. Review Comments to the Author

Reviewer #1: The authors have systematically and rightfully addressed all the issues in this revised version. This version of the paper may be accepted for publication.

Reviewer #2: Review Feedback

This abstract presents a comprehensive study examining the effects of financial metrics, free float shareholders, GDP growth, and firm age on dividend payout policy in both short- and long-term scenarios. The use of ARIMA and WLS models showcases methodological rigor, addressing common challenges in time-series data analysis. Key findings reveal that GDP growth negatively impacts long-term dividend payouts across industries, while factors like past dividends, debt-to-equity ratio, firm age, and free cash flows significantly influence short-term policies. The inclusion of half-life analysis highlights the study's practical implications and relevance for understanding dynamic financial behaviors.

However, the study writeup requires serious attention. Authors are suggested to proof read the writeup.

7. PLOS authors have the option to publish the peer review history of their article (what does this mean?). If published, this will include your full peer review and any attached files.

Reviewer #1: No

Reviewer #2: No

---

## [Editor Report · Acceptance letter]

2 Jan 2025

PONE-D-24-42377R1 

PLOS ONE

Dear Dr. Boonyanet, 

I'm pleased to inform you that your manuscript has been deemed suitable for publication in PLOS ONE. Congratulations! Your manuscript is now being handed over to our production team.

Kind regards, 

on behalf of

Dr. Michel Alexandre 

Academic Editor

PLOS ONE